**Data Availability Statement:** Data are available only upon request according to the "Act on the Protection of Personal Information" Law (as of

# Variation in in-hospital mortality and its association with percutaneous coronary intervention-related bleeding complications: A report from nationwide registry in Japan

Yuichi Sawayama[1], Kyohei Yamaji[2], Shun Kohsaka[3], Takashi Yamamoto[4], Yosuke Higo[1], Yohei Numasawa[5], Taku Inohara[3], Hideki Ishii[6], Tetsuya Amano[7], Yuji Ikari[8], Yoshihisa Nakagawa[1]*

1 Department of Cardiovascular Medicine, Shiga University of Medical Science, Otsu, Japan, 2 Division of Cardiology, Kokura Memorial Hospital, Kitakyushu, Japan, 3 Department of Cardiology, Keio University School of Medicine, Tokyo, Japan, 4 Department of Cardiovascular Medicine, Kohka Public Hospital, Kohka, Japan, 5 Department of Cardiology, Japanese Red Cross Ashikaga Hospital, Ashikaga, Japan, 6 Department of Cardiovascular Medicine, Gunma University Graduate School of Medicine, Maebashi, Japan, 7 Department of Cardiology, Aichi Medical University, Aichi, Japan, 8 Department of Cardiology, Tokai University School of Medicine, Isehara, Japan

* nkgw4413@belle.shiga-med.ac.jp

## Abstract

Large-scale registries have demonstrated that in-hospital mortality after percutaneous coronary intervention (PCI) varies widely across institutions. However, whether this variation is related to major procedural complications (e.g., bleeding) is unclear. In this study, institutional variation in in-hospital mortality and its association with PCI-related bleeding complications were investigated. We analyzed 388,866 procedures at 718 hospitals performed from 2017 to 2018, using data from a nationwide PCI registry in Japan. Hospitals were stratified into quintiles according to risk-adjusted in-hospital mortality (very low, low, medium, high, and very high). Incidence of bleeding complications, defined as procedure-related bleeding events that required a blood transfusion, and in-hospital mortality in patients who developed bleeding complications were calculated for each quintile. Overall, 4,048 (1.04%) in-hospital deaths and 1,535 (0.39%) bleeding complications occurred. Among patients with bleeding complications, 270 (17.6%) died during hospitalization. In-hospital mortality ranged from 0.22% to 2.46% in very low to very high mortality hospitals. The rate of bleeding complications varied modestly from 0.27% to 0.57% (odds ratio, 1.95; 95% confidence interval, 1.58–2.39). However, mortality after bleeding complications markedly increased by quintile and was 6-fold higher in very high mortality hospitals than very low mortality hospitals (29.0% vs. 4.8%; odds ratio, 12.2; 95% confidence interval, 6.90–21.7). In conclusion, institutional variation in in-hospital mortality after PCI was associated with procedure-related bleeding complications, and this variation was largely driven by differences in mortality after bleeding complications rather than difference in their incidence. These findings underscore the importance of efforts toward reducing not only bleeding complications but also, even more importantly, subsequent mortality once they have occurred.

May 2017) and the "Ethical Guidelines for Medical and Health Research Involving Human Subjects" (as of March 2015). The current study data were obtained from the J-PCI registry and would be available upon request to the University of Tokyo, Healthcare Quality Assessment, and Japanese Association of Cardiovascular Intervention and Therapeutics Registry Subcommittee (e-mail: info@cvit.jp).

**Funding:** The authors received no specific funding for this work.

**Competing interests:** I have read the journal's policy and the authors of this manuscript have the following competing interests: Dr. Yamaji has received investigator-initiated grant funding from Abbott. Dr. Kohsaka has received investigator-initiated grant funding from Bayer and Daiichi Sankyo and lecture fees from Bristol-Myers Squibb. Dr. Ishii has received lecture fees from Astellas Pharma, AstraZeneca, Bayer, Chugai Pharma Inc., Daiichi Sankyo, and MSD. Dr. Amano has received lecture fees from Astellas Pharma, AstraZeneca, Bayer, Daiichi Sankyo, and Bristol-Myers Squibb. Dr. Ikari has received research grants from Boston Scientific and Bayer. Dr. Nakagawa has received investigator-initiated grant funding from Terumo, Abbott, and Boston Scientific and lecture fees from Daiichi Sankyo, Bayer, and Bristol-Myers Squibb. The other authors declare no conflicts of interest associated with this manuscript. This does not alter our adherence to PLOS ONE policies on sharing data and materials.

## Introduction

Despite advances in percutaneous coronary intervention (PCI) over the last 40 years, bleeding has been regarded as one of the most serious procedure-related complications [1, 2]. Once bleeding complications occur, in-hospital mortality increases by approximately 12% [2]. The introduction of radial access has contributed to reductions in bleeding complications [3–5]; however, not only access site but also non-access site bleeding complications was independently associated with an increased risk of postprocedural mortality [6, 7]. Moreover, the recent advent of novel potent antithrombotic agents has increased bleeding events in exchange for a reduction in ischemic events [8, 9]. Especially recently, some populations receiving PCI comprise patients with a high risk of bleeding [10, 11]; therefore, PCI-related bleeding complications and their related mortality remain significant issues worldwide.

Large-scale registries have demonstrated that in-hospital mortality in patients underwent PCI varies widely across institutions [12–14], though the underlying reasons are not fully understood. We hypothesized that this variation may be involved in bleeding complications because their incidence or subsequent clinical outcomes can be largely dictated by hospital's capacity (e.g., the ability to prevent, expeditiously recognize or properly manage complications). In this study, therefore, we aimed to investigate the association between institutional variation in in-hospital mortality and bleeding complications within a representative nationwide PCI registry in Japan. Identifying this association has the potential to improve the prognosis of PCI in the contemporary era when patients with a high risk of bleeding are commonly treated.

## Material and methods

### Data source and study patients

The Japanese Percutaneous Coronary Intervention (J-PCI) registry is an ongoing nationwide PCI registry endorsed by the Japanese Association of Cardiovascular Intervention and Therapeutics (CVIT) that was designed to record clinical characteristics and in-hospital outcomes of patients undergoing PCI [15, 16]. In January 2013, the J-PCI registry was incorporated into the National Clinical Data system, a nationwide prospective Internet-based registry linked to board certification. Since all hospitals must participate in the J-PCI registry for board certification application and renewal, the degree of data completeness is high. Each hospital has a data manager who is responsible for collecting and recording PCI data. The CVIT holds an annual meeting of data managers to secure appropriate data collection and performs random audits (20 institutions annually) to check the quality of abstracted data. The definitions of variables in the J-PCI registries are available online from the CVIT. This study was conducted in accordance with the principles of the Declaration of Helsinki and approved by the Institutional Review Board of the Network for Promotion of Clinical Studies (a specialized nonprofit organization affiliated with Osaka University Graduate School of Medicine in Osaka, Japan). The requirement for written informed consent was waived because of the retrospective study design. In accordance with the Transparency and Openness Promotion Guidelines, the data that support the findings of this study are available from the corresponding author upon reasonable request.

This study population consisted of consecutive cases registered from January 2017 to December 2018 in the J-PCI registry. Then, cases with missing information in background characteristics were excluded. Also, we restricted the dataset to institutions that reported at least one in-hospital death during the study period (Fig 1).

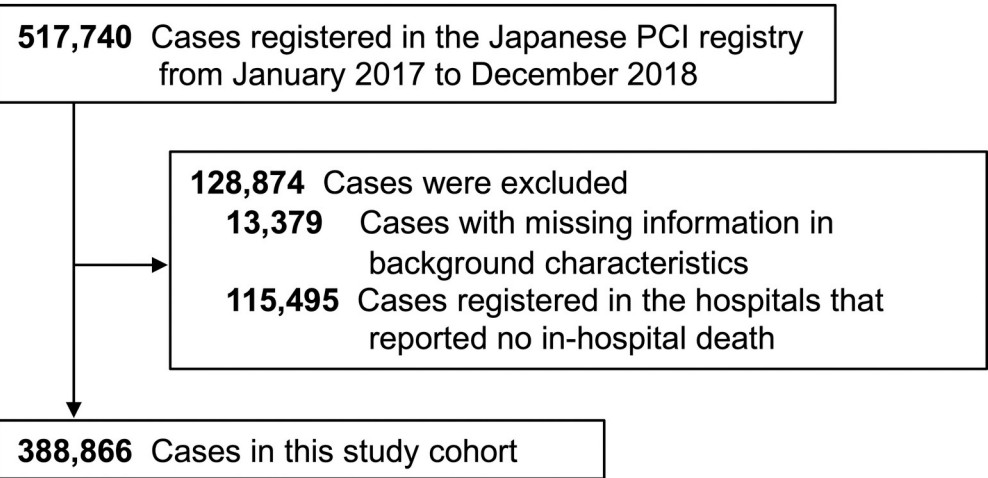

**Fig 1. Flowchart of study enrollment.** PCI indicates percutaneous coronary intervention.

## Definition of variables

Bleeding complication was defined as a PCI-related bleeding event that required a blood transfusion during the index PCI hospitalization. In-hospital mortality was defined as death from any cause. Other study variables, including patient characteristics, clinical presentation, angiographic and procedural details, and in-hospital outcomes were defined as previously reported [15].

## Model of risk adjusted in-hospital mortality

To account for differences in patient variables that affect mortality, hospitals were stratified into quintiles according to risk-adjusted in-hospital mortality, which was calculated as follows: (i) with reference to previous study [17], a multivariable logistic regression model was created to predict individual patient probability of death using age, sex, smoking within 1 year, hypertension, dyslipidemia, chronic kidney disease, maintenance dialysis, peripheral vascular disease, previous PCI, previous coronary artery bypass graft surgery, previous heart failure, cardiogenic shock within 24 hours, clinical presentation (ST-segment elevation myocardial infarction, non-ST-segment elevation myocardial infarction, unstable angina, or others), access site, number of diseased vessels, antiplatelet agents given at the time of PCI, and anticoagulant agents given at the time of PCI as explanatory variables (C-statistic of 0.91); (ii) crude mortality at each hospital was calculated and predicted probability of death for each patient at each hospital was averaged to obtain expected mortality; (iii) finally, risk-adjusted mortality at each hospital was calculated by dividing crude mortality by expected mortality and multiplying it by overall database mortality. We then ranked the hospitals according to risk-adjusted mortality and stratified them into quintiles (very low, low, medium, high, and very high mortality).

## Main analysis and subanalysis

As a main analysis, we calculated incidence of bleeding complications and mortality in patients who developed bleeding complications for each quintile. As a subanalysis, access and non-access site bleeding events were assessed individually. In addition, access site bleeding events were assessed separately for transfemoral access (TFA) and transradial access (TRA). The

incidence of TFA- and TRA-related bleeding events was calculated among only patients treated via TFA and TRA, respectively.

## Statistical analysis

Categorical data are reported as numbers with percentage and were compared using the chi-square test. Continuous data with normal distribution are expressed as means with standard deviation. Continuous data with non-normal distribution are expressed as medians with inter-quartile range. One-way analysis of variance and the Kruskal–Wallis test were used to compare continuous data. Incidence of bleeding complications and mortality in patients who developed bleeding complications are presented as numbers with percentage. Odds ratios (ORs) with 95% confidence interval (CI) were calculated in each quintile relative to very low mortality hospitals using mixed model logistic regression. Institutions were included in the models as random intercepts. The Cochran–Armitage test was used to assess trends in complication incidence and mortality after complications among the quintiles. Two-sided $P$ <0.05 was considered significant. All statistical analyses were performed using R software version 3.6.3 (R Foundation for Statistical Computing, Vienna, Austria).

## Results

A total of 388,866 cases at 718 hospitals were included for analysis. Overall, 4,048 (1.04%) in-hospital deaths occurred. Patient characteristics according to risk-adjusted mortality quintiles are summarized in Table 1. In-hospital mortality ranged from 0.22% to 2.46% in very low to very high mortality hospitals. Compared to other quintiles, the very low mortality quintile of hospitals had a 1.5-fold higher proportion of hospitals that performed >500 PCI cases per year (Table 2). TFA was used more frequently in the lower mortality hospitals, whereas TRA was used more frequently in the higher mortality hospitals. The prevalence of acute clinical presentation, including ST-segment elevation myocardial infarction and non-ST-segment elevation myocardial infarction, was highest in medium mortality hospitals. Complex PCI cases requiring rotational atherectomy or directional coronary atherectomy were performed predominantly in elective PCI cases and were more frequent in lower mortality hospitals.

Bleeding complications occurred in 1,535 cases overall (0.39%) and modestly increased from 0.27% in very low mortality hospitals to 0.57% in very high mortality hospitals (OR, 1.95; 95% CI, 1.58–2.39; Table 3A). Mortality in patients who developed bleeding complications is summarized in Table 3B. Of the 1,535 patients with overall bleeding complications, 270 (17.6%) died during hospitalization. Unlike the trend for incidence of bleeding complications, in-hospital mortality after bleeding complications increased markedly by quintile (Fig 2) and was 6-fold higher in very high mortality hospitals compared to very low mortality hospitals (29.0% vs. 4.8%; OR, 12.2; 95% CI, 6.90–21.7). Incidence of bleeding at the access site (0.21%) and non-access sites (0.19%) was similar. Incidence of bleeding complications at non-access sites tended to be higher in the higher mortality hospitals: the bleeding at non-access sites was 3 times higher in very high mortality hospitals than very low mortality hospitals (0.34% vs. 0.098%; OR, 3.17; 95% CI, 2.41–4.16). However, this tendency was weaker for access site bleeding complications (0.24% vs. 0.18%; OR, 1.33; 95% CI, 1.03–1.71; Table 3A). In-hospital mortality was twice as high after bleeding complications at non-access sites (25.4%) than after bleeding complications at the access site (11.1%). Mortality after bleeding complications differed greatly between very high and very low mortality hospitals for both bleeding complications at non-access sites (35.2% vs. 6.7%; OR, 17.7; 95% CI, 8.07–38.8) and the access site (20.9% vs. 3.7%; OR, 7.52; 95% CI, 3.29–17.2; Table 3B).

**Table 1. Patient characteristics.**

| | Overall | Very low mortality | Low mortality | Medium mortality | High mortality | Very high mortality | P value |
|---|---|---|---|---|---|---|---|
| Number of PCI cases | 388,866 | 106,591 | 73,323 | 69,096 | 71,206 | 68,650 | |
| In-hospital death | 4,048 (1.04) | 232 (0.22) | 419 (0.57) | 665 (0.96) | 1,040 (1.46) | 1,692 (2.46) | <0.001 |
| Age, years | 71 ± 11 | 70 ± 11 | 71 ± 11 | 70 ± 11 | 71 ± 11 | 70 ± 11 | <0.001 |
| Male | 297,464 (76) | 81,702 (77) | 56,083 (76) | 53,224 (77) | 54,043 (76) | 52,412 (76) | <0.001 |
| Smoking within 1 year | 119,873 (31) | 36,263 (34) | 20,373 (28) | 20,000 (29) | 23,310 (33) | 19,927 (29) | <0.001 |
| Diabetes | 173,147 (45) | 47,151 (44) | 32,592 (44) | 29,787 (43) | 32,723 (46) | 30,894 (45) | <0.001 |
| Hypertension | 291,399 (75) | 79,640 (75) | 54,403 (74) | 51,973 (75) | 54,306 (76) | 51,077 (74) | <0.001 |
| Dyslipidemia | 254,172 (65) | 69,010 (65) | 47,421 (65) | 45,134 (65) | 48,336 (68) | 44,271 (64) | <0.001 |
| Chronic kidney disease | 76,229 (20) | 20,493 (19) | 13,744 (19) | 13,029 (19) | 15,263 (21) | 13,700 (20) | <0.001 |
| Maintenance dialysis | 27,059 (7.0) | 7,692 (7.2) | 5,082 (6.9) | 4,533 (6.6) | 5,349 (7.5) | 4,403 (6.4) | <0.001 |
| COPD | 10,088 (2.6) | 2,627 (2.5) | 2,140 (2.9) | 1,572 (2.3) | 1,997 (2.8) | 1,752 (2.6) | <0.001 |
| Peripheral vascular disease | 30,292 (7.8) | 7,784 (7.3) | 5,710 (7.8) | 5,406 (7.8) | 5,824 (8.2) | 5,568 (8.1) | <0.001 |
| Previous PCI | 180,453 (46) | 51,093 (48) | 34,781 (47) | 31,119 (45) | 32,536 (46) | 30,924 (45) | <0.001 |
| Previous CABG | 14,148 (3.6) | 3,570 (3.3) | 2,709 (3.7) | 2,552 (3.7) | 2,970 (4.2) | 2,347 (3.4) | <0.001 |
| Previous heart failure | 57,042 (15) | 14,883 (14) | 11,486 (16) | 9,731 (14) | 11,132 (16) | 9,810 (14) | <0.001 |
| Cardiogenic shock within 24 hours | 13,419 (3.5) | 3,568 (3.3) | 2,710 (3.7) | 2,481 (3.6) | 2,513 (3.5) | 2,147 (3.1) | <0.001 |
| Baseline hemoglobin, g/dL | 13.2 ± 2.0 | 13.1 ± 2.0 | 13.2 ± 2.1 | 13.2 ± 2.0 | 13.2 ± 2.1 | 13.2 ± 2.1 | 0.002 |
| **Clinical presentation** | | | | | | | <0.001 |
| Acute setting | 148,715 (38) | 37,878 (36) | 27,998 (38) | 27,591 (40) | 28,476 (40) | 26,772 (39) | |
| STEMI | 68,910 (18) | 17,355 (16) | 12,755 (17) | 12,963 (19) | 12,799 (18) | 13,038 (19) | |
| NSTEMI | 21,217 (5.5) | 5,285 (5.0) | 3,779 (5.2) | 4,163 (6.0) | 4,047 (5.7) | 3,943 (5.7) | |
| UA | 58,591 (15) | 15,238 (14) | 11,467 (16) | 10,465 (15) | 11,630 (16) | 9,791 (14) | |
| **Access site** | | | | | | | <0.001 |
| Transfemoral | 98,625 (25) | 29,931 (28) | 20,493 (28) | 17,816 (26) | 15,694 (22) | 14,691 (21) | |
| Transradial | 270,838 (70) | 70,475 (66) | 49,230 (67) | 48,565 (70) | 51,593 (72) | 50,975 (74) | |
| **Number of diseased vessels** | | | | | | | |
| One | 237,110 (61) | 65,293 (61) | 45,205 (62) | 41,736 (60) | 43,912 (62) | 40,964 (60) | <0.001 |
| Two | 92,749 (24) | 25,375 (24) | 17,089 (23) | 16,868 (24) | 16,409 (23) | 17,008 (25) | <0.001 |
| Three | 42,796 (11) | 11,501 (11) | 8,070 (11) | 7,658 (11) | 7,672 (11) | 7,895 (12) | <0.001 |
| Left main | 16,211 (4.2) | 4,422 (4.1) | 2,959 (4.0) | 2,834 (4.1) | 3,213 (4.5) | 2,783 (4.1) | <0.001 |
| **Target vessel** | | | | | | | |
| RCA | 130,262 (33) | 35,714 (34) | 24,453 (33) | 22,857 (33) | 23,930 (34) | 23,308 (34) | 0.01 |
| LMCA and/or LAD | 204,398 (53) | 55,981 (53) | 37,876 (52) | 35,790 (52) | 38,159 (54) | 36,592 (53) | <0.001 |
| LCX | 95,475 (25) | 26,517 (25) | 18,043 (25) | 15,989 (23) | 17,849 (25) | 17,077 (25) | <0.001 |
| **Antithrombotic agents given at time of PCI** | | | | | | | |
| Antiplatelet agents | 357,447 (92) | 97,149 (91) | 65,072 (89) | 63,565 (92) | 67,774 (95) | 63,887 (93) | <0.001 |
| Aspirin | 345,703 (89) | 93,953 (88) | 62,893 (86) | 61,652 (89) | 65,597 (92) | 61,608 (90) | <0.001 |
| Clopidogrel | 135,698 (35) | 37,721 (35) | 24,565 (34) | 26,661 (39) | 24,292 (34) | 22,459 (33) | <0.001 |
| Prasugrel | 191,263 (49) | 50,054 (47) | 35,993 (49) | 31,079 (45) | 37,380 (52) | 36,757 (54) | <0.001 |
| Ticagrelor | 468 (0.1) | 219 (0.2) | 94 (0.1) | 68 (0.1) | 53 (0.1) | 34 (0.05) | <0.001 |
| Anticoagulant agents | 26,309 (6.8) | 6,694 (6.3) | 4,950 (6.8) | 4,962 (7.2) | 5,129 (7.2) | 4,574 (6.7) | <0.001 |
| Warfarin | 10,378 (2.7) | 2,793 (2.6) | 1,972 (2.7) | 1,805 (2.6) | 2,014 (2.8) | 1,794 (2.6) | <0.001 |
| Dabigatran | 1,410 (0.4) | 348 (0.3) | 260 (0.4) | 297 (0.4) | 236 (0.3) | 269 (0.4) | 0.01 |
| Rivaroxaban | 5,004 (1.3) | 1,326 (1.2) | 834 (1.1) | 907 (1.3) | 1,026 (1.4) | 911 (1.3) | <0.001 |
| Apixaban | 5,388 (1.4) | 1,260 (1.2) | 1,078 (1.5) | 1,110 (1.6) | 1,037 (1.5) | 903 (1.3) | <0.001 |
| Edoxaban | 4,128 (1.1) | 1,046 (1.0) | 824 (1.1) | 799 (1.2) | 795 (1.1) | 664 (1.0) | 0.051 |
| Dual antiplatelet therapy | 316,061 (81) | 84,998 (80) | 58,456 (80) | 55,841 (81) | 59,763 (84) | 57,003 (83) | <0.001 |

*(Continued)*

**Table 1.** (Continued)

| | Overall | Very low mortality | Low mortality | Medium mortality | High mortality | Very high mortality | P value |
|---|---|---|---|---|---|---|---|
| Triple therapy* | 19,205 (4.9) | 4,623 (4.3) | 3,727 (5.1) | 3,550 (5.1) | 3,951 (5.5) | 3,354 (4.9) | <0.001 |
| **Therapeutic devices** | | | | | | | |
| Balloon | 335,129 (86) | 95,165 (89) | 61,721 (84) | 60,228 (87) | 57,895 (81) | 60,120 (88) | <0.001 |
| BMS | 3,169 (0.81) | 764 (0.72) | 657 (0.90) | 527 (0.76) | 817 (1.1) | 404 (0.59) | <0.001 |
| DES | 329,590 (85) | 89,029 (84) | 61,966 (85) | 58,776 (85) | 60,893 (86) | 58,926 (86) | <0.001 |
| Rotational atherectomy | 15,772 (4.1) | 5,108 (4.8) | 2,733 (3.7) | 2,436 (3.5) | 2,906 (4.1) | 2,589 (3.8) | <0.001 |
| DCA | 2,256 (0.58) | 746 (0.70) | 619 (0.84) | 446 (0.65) | 198 (0.28) | 247 (0.36) | <0.001 |

Values are expressed as means ± standard deviation or numbers (%). Chronic kidney disease was defined as the presence of proteinuria, and/or a serum creatinine level ≥1.3 mg/dL, and/or an estimated glomerular filtration rate level ≤60 mL/min per 1.73 m$^2$.

BMS, bare metal stent; CABG, coronary artery bypass grafting; COPD, chronic obstructive pulmonary disease; DCA, directional coronary atherectomy; DES, drug-eluting stent; LAD, left anterior descending artery; LCX, left circumflex artery; LMCA, left main coronary artery; NSTEMI, non-ST-segment elevation myocardial infarction; PCI, percutaneous coronary intervention; RCA, right coronary artery; STEMI, ST-segment elevation myocardial infarction; UA, unstable angina.

*Triple therapy indicates an anticoagulant agent plus dual antiplatelet therapy.

TFA was associated with an approximately 10-fold higher incidence of bleeding complications than TRA (0.61% vs. 0.065%). As shown in Fig 3A, TFA-related bleeding complications tended to occur more frequently in higher mortality hospitals. Incidence of TFA-related bleeding in very high and very low mortality hospitals was 0.76% and 0.44%, respectively. In contrast, incidence of TRA-related bleeding complications was relatively similar across quintiles. Mortality after bleeding complications was identical in the TFA and TRA groups (12%). Consistent with the main analysis results, mortality after TFA-related bleeding differed greatly between very high and very low mortality hospitals (22.5% vs. 3.0%), but the trend was weaker in the TRA-related bleeding group (20.5% vs. 7.1%; Fig 3B). Therefore, among TFA patients, the incidence of bleeding complications and subsequent death increased by quintile. Incidence of bleeding and subsequent death for very high and very low mortality hospitals was 0.17% and 0.013%, respectively. This trend was also observed in TRA patients, but the absolute

**Table 2. Institutional characteristics.**

| | Overall | Very low mortality | Low mortality | Medium mortality | High mortality | Very high mortality | P value |
|---|---|---|---|---|---|---|---|
| Number of institutions | 718 | 144 | 143 | 144 | 143 | 144 | |
| Number of PCI cases | 452 (268–702) | 634 (461–875) | 429 (294–650) | 420 (244–630) | 398 (204–610) | 383 (210–672) | <0.001 |
| Number of institutions by PCI cases per year | | | | | | | <0.001 |
| <100 cases | 116 (16) | 1 (0.7) | 17 (12) | 28 (19) | 35 (24) | 35 (24) | |
| 100 to <500 cases | 534 (74) | 117 (81) | 117 (81) | 107 (74) | 95 (66) | 98 (68) | |
| 500 to <1000 cases | 60 (8.4) | 23 (16) | 8 (5.6) | 9 (6.3) | 10 (6.9) | 10 (6.9) | |
| ≥1000 cases | 8 (1.1) | 3 (2.1) | 1 (0.7) | 0 (0.0) | 3 (2.1) | 1 (0.7) | |
| In-hospital mortality, % | | | | | | | |
| Expected mortality | 1.0 (0.7–1.4) | 1.1 (0.8–1.3) | 1.1 (0.8–1.5) | 1.0 (0.8–1.4) | 1.0 (0.7–1.4) | 0.9 (1.6–1.2) | 0.004 |
| Crude mortality | 0.9 (0.4–1.7) | 0.2 (0.2–0.3) | 0.5 (0.4–0.8) | 1.0 (0.7–1.2) | 1.4 (1.0–2.0) | 2.4 (1.6–3.3) | <0.001 |
| Risk-adjusted mortality | 0.9 (0.5–1.7) | 0.2 (0.2–0.3) | 0.5 (0.5–0.6) | 0.9 (0.8–1.0) | 1.5 (1.3–1.7) | 2.6 (2.2–3.4) | <0.001 |

Values are expressed as medians (interquartile range) or numbers (%).

PCI, percutaneous coronary intervention.

**Table 3. Incidence of bleeding complications and in-hospital death in patients who developed bleeding complications.**

|  | Overall | Very low mortality | Low mortality | Medium mortality | High mortality | Very high mortality | P for trend |
|---|---|---|---|---|---|---|---|
| **(A) Incidence of bleeding complications** | | | | | | | |
| **Overall** | 1,535 (0.39) | 291 (0.27) | 245 (0.33) | 322 (0.47) | 284 (0.40) | 393 (0.57) | <0.001 |
|  |  | Reference | 1.25 (1.01–1.55) | 1.72 (1.39–2.12) | 1.57 (1.27–1.94) | 1.95 (1.58–2.39) |  |
| **Non-access site** | 744 (0.19) | 104 (0.098) | 98 (0.13) | 160 (0.23) | 149 (0.21) | 233 (0.34) | <0.001 |
|  |  | Reference | 1.36 (1.00–1.85) | 2.34 (1.76–3.11) | 2.25 (1.69–3.00) | 3.17 (2.41–4.16) |  |
| **Access site** | 829 (0.21) | 190 (0.18) | 161 (0.22) | 170 (0.25) | 145 (0.20) | 163 (0.24) | 0.02 |
|  |  | Reference | 1.28 (0.99–1.64) | 1.40 (1.09–1.80) | 1.21 (0.94–1.57) | 1.33 (1.03–1.71) |  |
| **(B) In-hospital death in patients who developed bleeding complications** | | | | | | | |
| **Overall** | 270 (17.6) | 14 (4.8) | 30 (12.2) | 52 (16.1) | 60 (21.1) | 114 (29.0) | <0.001 |
|  |  | Reference | 3.10 (1.62–5.94) | 5.63 (3.07–10.3) | 6.64 (3.65–12.1) | 12.2 (6.90–21.7) |  |
| **Non-access site** | 189 (25.4) | 7 (6.7) | 24 (24.4) | 35 (21.9) | 41 (27.5) | 82 (35.2) | <0.001 |
|  |  | Reference | 4.96 (2.11–11.7) | 7.61 (3.34–17.3) | 9.06 (4.01–20.5) | 17.7 (8.07–38.8) |  |
| **Access site** | 92 (11.1) | 7 (3.7) | 10 (6.2) | 19 (11.2) | 22 (15.2) | 34 (20.9) | <0.001 |
|  |  | Reference | 2.07 (0.78–5.49) | 4.15 (1.72–9.99) | 4.76 (2.01–11.3) | 7.52 (3.29–17.2) |  |

The upper row of each line shows the number of events (%) and the lower row shows the odds ratio (OR) with 95% confidence interval (CI) in each quintile (calculated relative to very low mortality hospitals).

number of events was extremely low. In these patients, incidence of bleeding and subsequent death for very high and very low mortality hospitals was 0.016% and 0.0043%, respectively (Fig 3C).

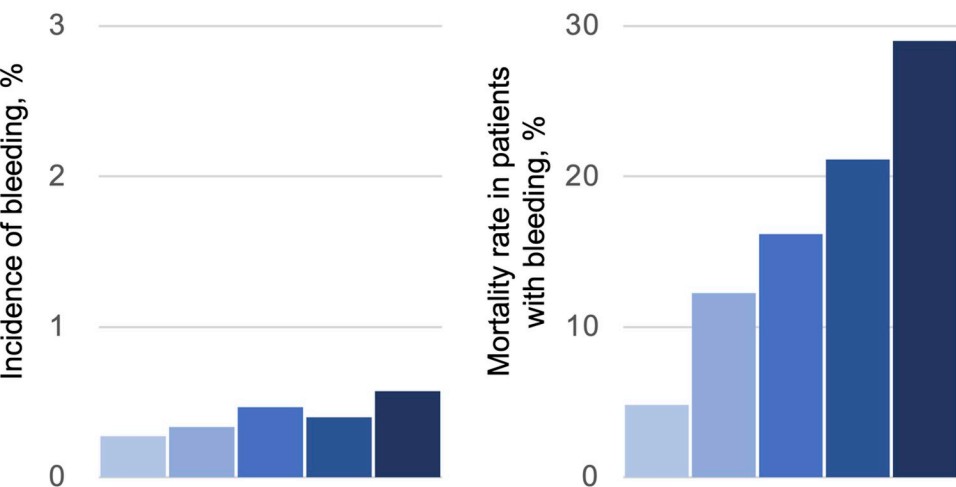

**Fig 2. Incidence of overall bleeding complications and in-hospital mortality in patients who developed bleeding complications.** When divided hospitals into quintiles according to their risk-adjusted mortality, the bleeding complication rate increased modestly from 0.27% to 0.57% in very low to very high mortality hospitals. However, the mortality rate in patients who developed bleeding complications markedly increased from 4.8% to 29.0% in very low to very high mortality hospitals.

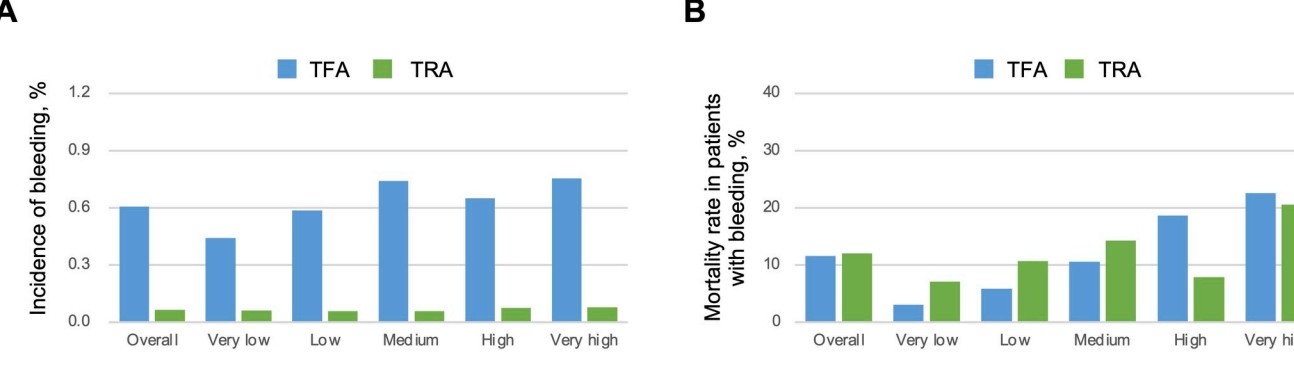

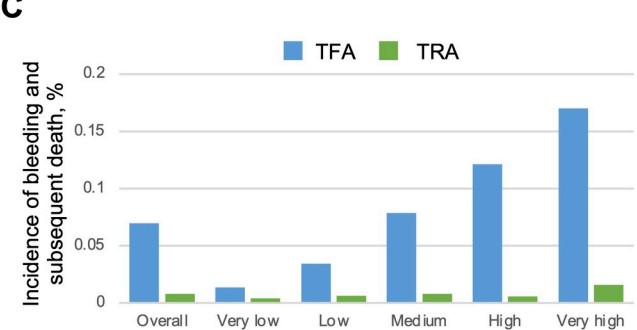

**Fig 3. Comparison between transfemoral and transradial access.** Hospitals were stratified into quintiles according to risk-adjusted mortality. The blue and green bars indicate the following rates for transfemoral access (TFA) and transradial access (TRA), respectively: (A) Incidence of bleeding complications. (B) Mortality in patients who developed bleeding complications. (C) Incidence of bleeding complications and subsequent in-hospital death.

## Discussion

This study examined 388,866 PCI procedures performed at 718 hospitals and registered within the J-PCI registry. Our main findings are as follows: (i) institutional variation in in-hospital mortality after PCI was associated with procedure-related bleeding complications; (ii) this variation was mainly caused by differences in mortality of patients who developed bleeding complications rather than crude incidence of bleeding complications; (iii) this trend was consistently observed in both bleeding complications from non-access and access site.

In-hospital mortality after PCI varies widely between hospitals [12–14]. Our results are in agreement, as in-hospital mortality in this study ranged from 0.22% to 2.46% in very low to very high mortality hospitals. Incidence of PCI-related bleeding events has also been reported to vary widely across institutions [18]. However, we found only modest variation across the studied hospitals. In contrast, mortality after bleeding complications varied significantly. Failure to rescue, defined as death in patients who develop a major procedural complication, is well established as an indicator of surgical quality of care [19]. In addition, institutional differences in failure-to-rescue rates underlie the wide variation in in-hospital mortality after surgery [20, 21]. The results obtained from our study were consistent with these reports, this concept may be applied to PCI (i.e., institutional differences in failure-to-rescue rates after bleeding complications underlie the wide variation in in-hospital mortality after PCI).

In this study, a tendency that variation in in-hospital mortality was mainly driven by difference in mortality after bleeding complications was consistently observed in both those from access and non-access site. Moreover, among access site-related bleeding complications, TRA

and TFA showed a similar trend. Previous studies revealed that TRA for PCI is associated with lower risk of complications than TFA [3–5]. Similarly, our study showed a 10-fold lower incidence of bleeding complications for TRA compared to TFA. However, given that higher mortality hospitals had higher mortality rate after bleeding complications even in patients treated via TRA, appropriate management after bleeding complications will be required regardless method of access. Of note, higher mortality hospitals used TRA more frequently. In those hospitals, TFA patients were more prone to develop bleeding complications as well as subsequent in-hospital death. Similarly, in the Minimizing Adverse Haemorrhagic Events by TRansradial Access Site and Systemic Implementation of angioX (MATRIX) trial, prognosis of TFA patients was worse in institutions with higher use of TRA. The incidence of net ischemic and bleeding events in TFA patients according to low, intermediate, and high use of TRA was 8.9%, 9.5%, and 17.1% [5]. The worse prognosis in institutions that more frequently use TRA might be caused by increased incidence of TFA-related adverse events.

This study has several important limitations. First, the definition of bleeding complications in the current study was different from standardized definitions, such as the definition from Thrombolysis in Myocardial Infarction trial [22], the Global Use of Strategies to Open Occluded Arteries trial [23], or the Bleeding Academic Research Consortium [24]. This may have underestimated the actual incidence of bleeding complications in this study. Second, we cannot determine the causality between bleeding complications and death during hospitalization. However, failure to rescue, defined by all-cause death after major perioperative complications, is well established as an indicator of the surgical quality of care [19]. The results in our study were equivalent to these reports, and this concept may be applied to PCI. Third, blood transfusion practices and thresholds vary between hospitals [25], which may have affected our results. Fourth, the number of variables in the J-PCI registry is limited and measured or unmeasured confounders were present. Although we used a logistic regression model to reduce potential confounding, we cannot completely eliminate this limitation. Finally, the quality of the database used in this study is a significant issue. However, data in the J-PCI registry are audited regularly to ensure accuracy.

In conclusion, institutional variation in in-hospital mortality after PCI was associated with procedure-related bleeding complications, and this variation was largely driven by differences in mortality after bleeding complications. These findings underscore the importance of efforts toward reducing not only bleeding complications but also, even more importantly, subsequent mortality once they have occurred. Further study is warranted on strategies for enabling improvement of clinical outcomes after bleeding complications.

## Acknowledgments

The authors appreciate the contributions of all the investigators and the members of the Japanese Association of Cardiovascular Intervention and Therapeutics for collecting data.

## Author Contributions

**Conceptualization:** Yuichi Sawayama, Yoshihisa Nakagawa.

**Data curation:** Kyohei Yamaji.

**Formal analysis:** Kyohei Yamaji.

**Methodology:** Yuichi Sawayama, Yoshihisa Nakagawa.

**Supervision:** Shun Kohsaka, Takashi Yamamoto, Taku Inohara, Hideki Ishii, Tetsuya Amano, Yuji Ikari, Yoshihisa Nakagawa.

Writing – **original draft:** Yuichi Sawayama.

Writing – **review & editing:** Kyohei Yamaji, Shun Kohsaka, Yosuke Higo, Yohei Numasawa, Hideki Ishii, Yoshihisa Nakagawa.

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
