## [Decision Letter · Decision Letter 0]

22 Oct 2021

PONE-D-21-20850Variation in in-hospital mortality and its association with percutaneous coronary intervention-related bleeding complications: A report from nationwide registry in JapanPLOS ONE

Dear Dr. Nakagawa,

Thank you for submitting your manuscript to PLOS ONE. After careful consideration, we feel that it has merit but does not fully meet PLOS ONE’s publication criteria as it currently stands. Therefore, we invite you to submit a revised version of the manuscript that addresses the points raised during the review process.

The manuscript is interesting but will require minor revisions.<o:p></o:p>

While they recognize the potential interest of the subject studied, the Reviewers raised some concerns that need to be properly addressed.<o:p></o:p>

We look forward to receiving your revised manuscript.

Kind regards,

Marcelo Arruda Nakazone, M.D., Ph.D.

Academic Editor

PLOS ONE

Journal Requirements:

[I have read the journal's policy and the authors of this manuscript have the following competing interests: Dr. Yamaji has received investigator-initiated grant funding from Abbott. Dr. Kohsaka has received investigator-initiated grant funding from Bayer and Daiichi Sankyo and lecture fees from Bristol-Myers Squibb. Dr. Ishii has received lecture fees from Astellas Pharma, AstraZeneca, Bayer, Chugai Pharma Inc., Daiichi Sankyo, and MSD. Dr. Amano has received lecture fees from Astellas Pharma, AstraZeneca, Bayer, Daiichi Sankyo, and Bristol-Myers Squibb. Dr. Ikari has received research grants from Boston Scientific and Bayer. Dr. Nakagawa has received investigator-initiated grant funding from Terumo, Abbott, and Boston Scientific and lecture fees from Daiichi Sankyo, Bayer, and Bristol-Myers Squibb. The other authors declare no conflicts of interest associated with this manuscript.]

Reviewers' comments:

Reviewer's Responses to Questions

**Comments to the Author**

1. Is the manuscript technically sound, and do the data support the conclusions?

Reviewer #1: Yes

Reviewer #2: Yes

2. Has the statistical analysis been performed appropriately and rigorously? 

Reviewer #1: I Don't Know

Reviewer #2: Yes

3. Have the authors made all data underlying the findings in their manuscript fully available?

Reviewer #1: Yes

Reviewer #2: No

4. Is the manuscript presented in an intelligible fashion and written in standard English?

Reviewer #1: Yes

Reviewer #2: Yes

5. Review Comments to the Author

Reviewer #1: Sawayama and colleagues have demonstrated a prognostic value of PCI-related bleeding complications. The study is of interest and timely, albeit with the inherent limitations of an observational retrospective study, as the need for more studies on the role of PCI-related bleeding is needed.

I have a few questions/clarifications that will be helpful in making it a stronger study:

1) The definition of bleeding complications in the current study was different from standardized definitions as the authors highlighted, but it is not clear how many patients had a fatal bleeding.

2) Haemoglobin and platelet values are not reported as the definition of Chronic Kidney Disease (eFGR < 60 ?).

3) It is not reported what type of intensive care unit (ICU) was present (cardiac surgery, neurosurgery) in the different hospitals: different ICUs means different types of treatment.

4) I think it is important to know what kind of antiplatelet agents and what kind of anticoagulant agents were used; moreover, how many patients were in triple (DAPT + anticoagulant) antithrombotic therapy?

5) It is not reported the use of intravascular imaging (IVUS, OCT).

6) It would be interesting to know if and which mechanical support (IABP, ECMO, Impella) was used in cardiogenic shock patients and how it could affect bleeding complications.

Reviewer #2: This paper reports results from the Japanese national PCI registry on bleeding complications from 388,866 procedures performed at 718 hospitals between 2017-2018. The authors found substantial variability in bleeding complications and mortality among the centers and provided some insights into associations of complications with center characteristics.

The paper adds valuable information on complications from PCI in contemporary practice. The authors acknowledge the study limitations, i.e., observational study design, which limits conclusions on causal relationships. Foremost, the authors should be careful ascribing all issues to bleeding complications since the latter, of course, may just be a result from vascular injury not from excessive anticoagulation. The authors do address this to some extent by differentiating access site vs. other complications but nevertheless, further information is needed to understand the nature of complications leading to bleeding.

6. PLOS authors have the option to publish the peer review history of their article (what does this mean?). If published, this will include your full peer review and any attached files.

Reviewer #1: No

Reviewer #2: No

---

## [Author Response · Author response to Decision Letter 0]

9 Nov 2021

Response to the Reviewer #1

1) The definition of bleeding complications in the current study was different from standardized definitions as the authors highlighted, but it is not clear how many patients had a fatal bleeding.

Response

We appreciate the comment from the reviewer. The J-PCI registry’s definition of bleeding complications is largely equivalent to Bleeding Academic Research Consortium (BARC) 3A-C or above. Previous studies demonstrated that the ratio of fatal bleeding among patients with BARC 3A-C or above was approximately 10% (Vranckx P, et al. J Am Coll Cardiol 2016, Ratcovich H, et al. Am J Cardiol 2021, Gargiulo G, et al. J Am Coll Cardiol 2018). Unfortunately, direct information on fatal bleeding was not available on the registry, but we believe the rate of fatal bleeding would be roughly the same for our study. We acknowledge that the causality between bleeding complications and in-hospital death cannot be proven; thus, we have added the descriptions in the Discussion section:

2) Haemoglobin and platelet values are not reported as the definition of Chronic Kidney Disease (eFGR < 60 ?).

Response

We had no data for platelet values in the J-PCI registry, but have added hemoglobin levels in Table 1 in the revised manuscript. CKD was defined as the presence of proteinuria, and/or a serum creatinine level ≥1.3 mg/dL, and/or an estimated glomerular filtration rate level ≤60 mL/min/1.73 m2. Based on these points, we have modified the descriptions in the Results section.

3) It is not reported what type of intensive care unit (ICU) was present (cardiac surgery, neurosurgery) in the different hospitals: different ICUs means different types of treatment.

Response

We agree it is also important in dealing with clinical outcomes such as in-hospital death. In Japan, the ICU is a place to treat patients with critical acute dysfunction, regardless of internal medicine or surgery. According to the report from Ministry of Health, Labour and Welfare in 2017 (https://www.e-stat.go.jp/dbview?sid=0003289748), the number of hospitals advocating ICU, coronary or cardiac care unit (CCU), and stroke care unit (SCU) is 713, 287, and 162, respectively. Other high care units, such as neurosurgical care unit, are practically very few and unreported. Some hospitals have multiple care units (e.g., ICU and SCU, etc), whereas others only have an ICU. Since the name varies from hospital to hospital, the J-PCI registry collects data under the name ICU. There is no data, but it is assumed that most of them mean CCU. 

4) I think it is important to know what kind of antiplatelet agents and what kind of anticoagulant agents were used; moreover, how many patients were in triple (DAPT + anticoagulant) antithrombotic therapy?

Response

We agree it is important in addressing the clinical outcome of bleeding complications, and we have added the information about status of antithrombotic therapy in Table 1 in the revised manuscript.

5) It is not reported the use of intravascular imaging (IVUS, OCT).

Response

We have no data for imaging devices for PCI in the J-PCI registry, but referring to studies from Japan in the same period (Watanabe H, et al. JAMA 2019, Nakamura M, et al. Circ J 2020), it is estimated that imaging devices were used in more than 80% of PCI cases during this period. Moreover, the real-world use of IVUS was associated with reduction in coronary dissection, but not bleeding complications (Kuno T, et al. Heart an Vessels 2019). Therefore, the impact of the use of imaging devices on bleeding complications or subsequent fatal events may not be significant.

6) It would be interesting to know if and which mechanical support (IABP, ECMO, Impella) was used in cardiogenic shock patients and how it could affect bleeding complications.

Response

We also recognize that the data on the use of mechanical support against cardiogenic shock would help us further understand bleeding complications or subsequent fatal events. However, the J-PCI registry during this study periods does not have sufficient data on these mechanical devices for cardiogenic shock (the registry became mandatory in the middle of 2018).

Response to the Reviewer #2

Reviewer #2

This paper reports results from the Japanese national PCI registry on bleeding complications from 388,866 procedures performed at 718 hospitals between 2017-2018. The authors found substantial variability in bleeding complications and mortality among the centers and provided some insights into associations of complications with center characteristics. The paper adds valuable information on complications from PCI in contemporary practice. The authors acknowledge the study limitations, i.e., observational study design, which limits conclusions on causal relationships. Foremost, the authors should be careful ascribing all issues to bleeding complications since the latter, of course, may just be a result from vascular injury not from excessive anticoagulation. The authors do address this to some extent by differentiating access site vs. other complications but nevertheless, further information is needed to understand the nature of complications leading to bleeding.

Response

We appreciate the Reviewer’s helpful comments and agree that we should be careful about ascribing all issues to bleeding complications. We acknowledge that the causality between bleeding complications and in-hospital death cannot be proven. First of all, we had no data to what extent bleeding complications were fatal. The J-PCI registry’s definition of bleeding complications is largely equivalent to Bleeding Academic Research Consortium (BARC) 3A-C or above. Previous studies demonstrated that the ratio of fatal bleeding among patients with BARC 3A-C or above was approximately 10% (Vranckx P, et al. J Am Coll Cardiol 2016, Ratcovich H, et al. Am J Cardiol 2021, Gargiulo G, et al. J Am Coll Cardiol 2018). Unfortunately, direct information on fatal bleeding was not available on the registry, but we believe the rate of fatal bleeding would be roughly the same for our study. Second, failure to rescue, defined by all-cause death after major perioperative complications, is well established as an indicator of the surgical quality of care (Silber JH, et al. Med Care 1992). Moreover, institutional differences in failure-to-rescue rates underlie the wide variation in in-hospital mortality after surgery (Ghaferi AA, et al. N Engl J Med 2009). We think that the results in our study were equivalent to these reports, and this concept may be applied to PCI. Third, hemoglobin level and status of antithrombotic agent use (i.e., whether DAPT or Triple therapy) at the time of index PCI are crucial factors when considering bleeding complications. Based on these points, we have added the descriptions in the Results and Discussion sections.

---

## [Decision Letter · Decision Letter 1]

1 Dec 2021

Variation in in-hospital mortality and its association with percutaneous coronary intervention-related bleeding complications: A report from nationwide registry in Japan

PONE-D-21-20850R1

Dear Dr. Nakagawa,

We’re pleased to inform you that your manuscript has been judged scientifically suitable for publication and will be formally accepted for publication once it meets all outstanding technical requirements.

Kind regards,

Marcelo Arruda Nakazone, M.D., Ph.D.

Academic Editor

PLOS ONE

Additional Editor Comments (optional):

Reviewers' comments:

Reviewer's Responses to Questions

**Comments to the Author**

1. If the authors have adequately addressed your comments raised in a previous round of review and you feel that this manuscript is now acceptable for publication, you may indicate that here to bypass the “Comments to the Author” section, enter your conflict of interest statement in the “Confidential to Editor” section, and submit your "Accept" recommendation.

Reviewer #1: All comments have been addressed

Reviewer #2: All comments have been addressed

2. Is the manuscript technically sound, and do the data support the conclusions?

Reviewer #1: Yes

Reviewer #2: Yes

3. Has the statistical analysis been performed appropriately and rigorously? 

Reviewer #1: Yes

Reviewer #2: Yes

4. Have the authors made all data underlying the findings in their manuscript fully available?

Reviewer #1: Yes

Reviewer #2: No

5. Is the manuscript presented in an intelligible fashion and written in standard English?

Reviewer #1: Yes

Reviewer #2: Yes

6. Review Comments to the Author

Reviewer #1: Well done!

I think prognostic value of PCI-related bleeding complications remains a "hot topic" and Japanese national PCI registry

could be very useful in the future.

Reviewer #2: The authors' responses are satisfactory and added explanations as well as additional text acknowledging limitations.

7. PLOS authors have the option to publish the peer review history of their article (what does this mean?). If published, this will include your full peer review and any attached files.

Reviewer #1: No

Reviewer #2: No

---

## [Editor Report · Acceptance letter]

3 Dec 2021

PONE-D-21-20850R1 

Variation in in-hospital mortality and its association with percutaneous coronary intervention-related bleeding complications: A report from nationwide registry in Japan 

Dear Dr. Nakagawa:

I'm pleased to inform you that your manuscript has been deemed suitable for publication in PLOS ONE. Congratulations! Your manuscript is now with our production department. 

Kind regards, 

on behalf of

Professor Marcelo Arruda Nakazone 

Academic Editor

PLOS ONE